# Buprenorphine Use for Analgesia in Palliative Care

**DOI:** 10.3390/pharmacy12030078

**Published:** 2024-05-13

**Authors:** Leanne K. Jackson, Ivy O. Poon, Mary A. Garcia, Syed Imam, Ursula K. Braun

**Affiliations:** 1Rehabilitation & Extended Care Line, Section of Palliative Medicine, Michael E DeBakey Veterans Affairs Medical Center, Houston, TX 77030, USA; maryacelle.garcia@va.gov (M.A.G.); syed.imam@va.gov (S.I.); ursula.braun@va.gov (U.K.B.); 2Division of Geriatric and Palliative Medicine, Baylor College of Medicine, Houston, TX 77030, USA; 3Pharmacy Practice, Texas Southern University, Houston, TX 77004, USA; ivy.poon@tsu.edu; 4Clinical Pharmacy Specialist/Pain & Palliative Care, Houston Methodist Hospital, Houston, TX 77030, USA

**Keywords:** buprenorphine, palliative care, cancer pain, pain, pain management, analgesics, narcotics, partial mu-opioid receptor

## Abstract

Buprenorphine is a semi-synthetic long-acting partial µ-opioid receptor (MOR) agonist that can be used for chronic pain as a sublingual tablet, transdermal patch (Butrans^®^), or a buccal film (Belbuca^®^). Buprenorphine’s unique high receptor binding affinity and slow dissociation at the MOR allow for effective analgesia while offering less adverse effects compared to a full agonist opioid, in particular, less concern for respiratory depression and constipation. It is underused in chronic pain and palliative care due to misconceptions and stigma from its use in opioid use disorder (OUD). This case report discusses the unique pharmacology of buprenorphine, including its advantages, disadvantages, available formulations, drug–drug interactions, initiation and conversion strategies, and identifies ideal populations for use, especially within the palliative care patient population.

## 1. Case Example

A 70-year-old edentulous Caucasian man with p16+ oropharyngeal squamous cell carcinoma (SCC) of the true vocal fold had undergone total laryngectomy ten years prior. He was found to have biopsy-proven SCC of the right tongue base. Two months of definitive chemotherapy and radiation with five cycles of weekly Carboplatin and Paclitaxel, along with 35 fractions of radiation, were completed. Scans and examination by Otolaryngology documented “no evidence of disease”. His past medical history was pertinent for chronic obstructive pulmonary disease (COPD), GOLD 2B on 2L of oxygen with exertion, depression, and hypothyroidism. All nutrition was provided by a gastrostomy tube. He had known upper esophageal stricture for which dilation had been attempted in the past. The patient had a 35-pack-year smoking history but had quit 15 years prior. A palliative care physician had been treating the patient during his cancer treatment and was working with the patient to wean his opioid requirements. His outpatient pain regimen included: fentanyl patch 25 mcg/h every 72 h (approximately 50 mg oral morphine equivalent, OME) and 30 mg of oral morphine equivalent for breakthrough dosing, for a total of 80 mg OME. Pain was present in the right ear, right jaw, right neck, and base of the skull. Abdominal cramping was also present. He presented with severe, multifactorial gastroparesis that necessitated tapering off more constipating opioids and repletion with intravenous thyroid medication, given poor absorption of parenteral medications. Given that his pain was not well controlled, his fentanyl patch was removed, and he was immediately started on 1 mg of buprenorphine/naltrexone sublingual tablets every 12 h. The short-acting morphine was continued for 4 days and then stopped. An additional 1 mg buprenorphine/naltrexone tablet once daily as needed was available as the bridge off of the full MOR agonist. A benefit of the fentanyl patch is the drug level declining slowly after patch removal, falling by approximately 50% in the subsequent 20 to 27 h, allowing for initiation and up-titration of buprenorphine without withdrawal. The patient achieved adequate analgesia on his new regimen.

## 2. Background

Buprenorphine is a semi-synthetic long-acting Schedule III partial MOR agonist [1]. It was first approved as a parenteral opioid by the Food and Drug Administration (FDA) in 1981. A sublingual agent and extended-release subcutaneous weekly or monthly formulation for opioid use disorder (OUD) were approved later. Buprenorphine also exists in a combination formulation with naloxone to discourage misuse. In the 2000s, a transdermal formulation (Butrans^®^) and then a buccal film (Belbuca^®^) were approved as long-term, around-the-clock, daily-use chronic pain treatments [2].

Buprenorphine’s unique receptor binding profile confers a range of benefits to effectively treat pain while offering a favorable adverse effect profile compared to a full agonist opioid. The high binding affinity and slow dissociation at the MOR allow for effective analgesia, while partial MOR agonism results in less respiratory depression. As an inverse agonist at the kappa opioid receptor (KOR) and antagonist at the delta receptor, buprenorphine can also result in less sedation, constipation, OUD, and cognitive impairment than traditional opioids [3].

The US Department of Health and Human Services Chronic Pain guideline in 2019 encouraged the use of buprenorphine as a first-line opioid for chronic pain [4]. Unfortunately, there has been hesitation to embrace the use of buprenorphine in chronic pain and palliative care due to misconceptions and stigma from its use in OUD. There remains confusion about the Drug Enforcement Administration (DEA) practitioner “X” waiver, misinterpretation about buprenorphine having a ceiling effect for analgesia, and fear of precipitating withdrawal when transitioning from full MOR to buprenorphine.

Here we will discuss the unique pharmacology of buprenorphine and identify ideal populations for use, including the population of patients in palliative care.

## 3. Pharmacology

### 3.1. Unique Aspects: Advantages

Buprenorphine is primarily excreted in feces and is thus safe for use in renal impairment.

It is a partial MOR agonist. MOR binds and activates both beta-arrestin and G proteins. Buprenorphine’s potent analgesia is achieved via G protein activation with high binding affinity and slow dissociation. The “partial” agonism of buprenorphine does not clinically equate to partial analgesia. In fact, combined with slower dissociation from MOR, buprenorphine has a lower potential for withdrawal and prolonged analgesia. Less binding on gastrointestinal MOR reduces constipation. The less beta-arrestin signaling allows for less side effects, better tolerability, and an improved safety profile when comparing buprenorphine to full MOR opioids. See Table 1 [3].

At doses of less than 16 mg buprenorphine per day, full analgesia is often achieved without all MOR being occupied. In addition, over time, buprenorphine can increase MOR expression; thus, it is still possible to use breakthrough opioids for acute pain crises while on long-term therapy with buprenorphine. KOR modulates stress, mood, reward, and pain through serotonin and dopamine neurotransmission. See Table 1 [3].

The combination of less beta-arrestin signaling and KOR antagonism leads to buprenorphine’s “antihyperalgesic” effects.

Full agonism of the opioid receptor like-1 (ORL1) blocks the rewarding actions and counteracts the antinociceptive actions of morphine, thus reducing reward and limiting tolerance. In addition, buprenorphine has preferential action on spinal receptors as opposed to higher central nervous system (CNS) receptors, and this is believed to decrease respiratory effects and reduce euphoria [3].

Buprenorphine does not inhibit serotonin reuptake, unlike methadone, fentanyl, and tramadol, and therefore does not contribute to possible serotonin syndrome [5].

Compared to methadone, buprenorphine has less of an effect on QTc prolongation [3]. Buprenorphine is metabolized via the CYP3A4 pathway, whereas methadone is metabolized through three CYP3A4, CYP2B6, and CYP2D6 pathways [5,6]. The half-lives of IM, buccal, and transdermal buprenorphine are 2 to 3 h, 27 h, and 26 h, respectively [7,8].

### 3.2. Unique Aspects: Disadvantages

Cannabis can increase serum buprenorphine levels; thus, patients should be counseled on the risk of withdrawal symptoms if cannabis is stopped or the risk of intoxication with new cannabis use [5,9]. In addition, one study found that the metabolite-to-parent ratio is higher in non-users (0.98) than in cannabis users (0.38), *p* = 0.02, thus indicating that cannabis use decreases norbuprenorphine formation, while elevating blood concentration of buprenorphine and norbuprenorphine, likely due to inhibition of CYP3A4. Physicians and pharmacists should inform patients about this pharmacokinetic interaction [9].

The FDA has issued a warning that sublingual (SL) administration of buprenorphine can result in tooth decay, cavities, loss of teeth, and oral infections, even in patients with no prior dental issues. A study of 6616 new SL buprenorphine/naloxone users was compared to 5385 new transdermal (TD) buprenorphine users, rates of any adverse dental events were 21.6 per 1000 person-years in the SL group and 12.2 per 1000 person-years in the TD group. Thus, SL buprenorphine confers a hazard ratio (HR) of 1.7 when compared to TD buprenorphine. The SL formulation is acidic, and administration of this formulation requires the medication to be held under the tongue for 5 to 10 min. Patients should be encouraged to obtain regular oral health examinations by their dentist [10].

While safe to use in mild-to-moderate hepatic impairment, a combination of buprenorphine and naloxone should be avoided in severe hepatic impairment because of increased naloxone concentrations. Patients should be cautioned to rotate the site of application of transdermal buprenorphine, or risk increased absorption if applied to the same site.

Buprenorphine overdoses require high-dose continuous intravenous naloxone 2–4 mg over 90 min to reverse and can therefore not easily be treated with nasal naloxone at home. Caregivers should be counseled to call 911 immediately after administration if nasal naloxone is required at home [3].

## 4. Formulations

Buprenorphine is available in multiple dosage forms, including injection solution, buccal film, transdermal patch, subcutaneous injection, and sublingual tablets [7]. Table 2 shows the details of each formulation available in the US. Among the formulations, injection solution, buccal film, and transdermal patch are FDA-approved for pain management, whereas the rest of the formulations are approved for opioid use disorder [11].

The buccal film and transdermal patch formulations are used for moderate to severe chronic pain due to a slower onset of action, longer elimination half-life, and the duration of action. When compared to the IV administration, IM, buccal film, and transdermal formulations have bioavailabilities of 70%, 46–65%, and 15%, respectively. Absorption of the drug may be increased when a patient has an elevated body temperature or applies a heating pad onto the transdermal system; reports have shown blood concentrations of buprenorphine increased by 26% to 55% when a heating pad is used [12,13,14]. Human skin in vitro permeation tests (IVPT) at 32 +/− 1 degree Celsius or 42 +/− 1 degree Celsius, have showed heat-induced enhancement in flux and the cumulative amount of drug permeated from transdermal buprenorphine. Thus, patients should be counseled to avoid heating pads, and to notify their physician if they experience high fever [13]. Ingestion of liquids decreases systemic exposure to buprenorphine from buccal film by 23% to 37%, and patients should refrain from eating and drinking for 30 min after application or until the buccal film is completely dissolved [11].

Buprenorphine is highly protein-bound to alpha and beta globulin by up to 96%. It is primarily metabolized hepatically via N-dealkylation through CYP3A4 to norbuprenorphine (active metabolite), and to a lesser extent via glucuronidation by UGT1A1 and 2B7 to buprenorphine 3-O-glucuronide. The major metabolite, norbuprenorphine, also undergoes glucuronidation via UGT1A3. Studies have shown that a significant proportion of sublingual buprenorphine is being swallowed and is thus subject to first-pass metabolism, whereas those who take parenteral formulations have been reported to experience less drug–drug interaction associated with CYP3A4 inhibitors or inducers [15]. Common CYP3A4 inhibitors and inducers are listed in Table 3. When a patient is initiated on buprenorphine, it is recommended to collect baseline liver function tests, including AST, ALT, and alkaline phosphatase (AKP), to make sure the values are less than five times the upper normal limits for AST/ALT, or three times the upper normal limits for AKP [16]. Additionally, when a CYP3A4 inhibitor or inducer is administered concurrently with buprenorphine, the drug level of buprenorphine may change, and additional monitoring of pain relief, respiration, symptoms of withdrawal, overdose, CNS depression, and liver function tests (LFT) may be necessary [7].

Buprenorphine is typically not listed in the morphine milligram equivalent (MME) table in the clinical practice guidelines and opioid equivalency table [18]. The CDC Clinical Practice Guideline for Prescribing Opioids for Pain excluded buprenorphine from the MME table because of the lower likelihood of abuse potential of buprenorphine compared to other full opioid agonists: “Buprenorphine should not be counted in the total MME/day in calculations because of its partial agonist properties at opioid receptors that confer a ceiling effect on respiratory depression” [18,19,20]. The manufacturer package insert of transdermal buprenorphine suggests using a lower starting dose of 5 mcg/h for those who are receiving MME of <30 mg per day and 10 mcg/h for those taking MME of ≥30 mg per day [12]. For the buccal film buprenorphine, the recommended starting doses are 75 mcg, 150 mcg, and 300 mcg every 12 h for those taking <30 mg MME, 30–89 mg, and 90–160 mg, respectively [11]. When converting from another opioid agonist, the manufacturer also suggests tapering down the patient’s current opioid to ≤30 mg oral MME prior to initiating buprenorphine to prevent withdrawal, which can be impractical for patients with chronic pain. Alternatively, an expert panel of pain management experts recommended the following direct conversion strategies [8], see Table 4.

A Dutch guideline on pain management in patients with cancer provided an opioid conversion table with a 52.5 mcg/h buprenorphine patch equivalent to 120 mg of oral morphine, and 105 mcg/h buprenorphine patch estimated to be equivalent to 240 mg of oral morphine [21].

## 5. Initiation Methods

Microdosing strategies are used to avoid withdrawal when converting from high-dose scheduled II opioids to buprenorphine. One method is to use 1 mg sublingual or 500 µg intravenously on day one while maintaining the dose of schedule II opioid. The buprenorphine dose is doubled daily until ~12 mg when the schedule II opioid can be discontinued without withdrawal. With its high affinity, buprenorphine blocks the schedule II opioid at the MOR [5].

Transdermal buprenorphine is useful when converting from a potent opioid because the slow rise in blood levels with a simultaneous slow decline in the level of potent opioids prevents withdrawal from occurring.

The “stop-start” approach for patients already receiving other opioids is not recommended for the palliative care population where consistent analgesia is the goal. “Stop-start” is commonly used for OUD and employs a “gap” approach in which patients stop all opioids prior to induction therapy with buprenorphine. For non-methadone opioids the “gap” time is 12–24 h without opioids, and for methadone it is 48–72 h of methadone abstinence. If the “stop-start” method is employed, transdermal buprenorphine could be used during the gap time to provide some pain control while up-titrating oral or SL buprenorphine formulations [3]. In the palliative care populations or the chronic pain population, microdosing strategies or the expert panel recommendations in Table 4 could be used for conversion to buprenorphine.

## 6. Ideal Populations for Use

Tolerance, opioid dependence, hyperalgesia, allodynia, sleep-disordered breathing, dysphoria, and a risk of developing OUD are all risks of long-term opioid therapy. For some patients, increasingly prolonged periods of opioid use are required due to (1) palliative care being offered earlier in the disease trajectory, (2) more patients surviving longer due to the beneficial effect of novel therapies, (3) patients who are cured of their cancer but continue to have severe pain, and (4) patients with pre-existing OUD who require opioid therapy for their illness [22]. The prevalence of OUD in patients with cancer was reported at 8% and the risk of developing OUD with traditional opioids was 23.5% [23]. Opioid tapers with buprenorphine allow for decreased withdrawal symptoms [3].

The combination of opioids with gabapentinoids and sedative-hypnotics, including benzodiazepines and barbiturates is known to increase the risk of mortality due to lethal respiratory depression [24]. Anxiety is common in patients receiving palliative care. With its ceiling effect on respiratory depression, buprenorphine might be a safer alternative for those requiring these adjunct medications. In addition, patients who consume alcohol, sedating antidepressants, antipsychotics, or any other concomitant central nervous system drugs associated with hypoventilation might be good buprenorphine candidates [25].

In older adults, buprenorphine is considered the safest opioid for chronic pain and thus should be considered as a first-line agent in the palliative care setting [25]. Lower risk of falls, less sedation and cognitive impairment are advantages in the aging population [3]. Older adults commonly have impaired renal function; thus, all opioids, except buprenorphine, would require longer time intervals, reduced doses, and closer monitoring of creatinine clearance [25]. Hepatic impairment might also be present in geriatric patients. Immunosuppression might be exacerbated by traditional opioids (in particular morphine and fentanyl); however, buprenorphine has not been observed to have an immunosuppressive effect [26]. Table 5 summarizes patient groups who may be good candidates for buprenorphine.

## 7. Barriers to Use

### 7.1. Former Regulatory Barrier in the United States

On 29 December 2022, President Biden signed the Consolidated Appropriates Act of 2023, which included the Mainstreaming Addiction Treatment Act (MAT Act). Before the MAT, the United States DEA required a special DEA certification, known as an “X-waiver” to prescribe buprenorphine for the treatment of opioid use disorder. This Drug Addiction Treatment ACT of 2000 (DATA 2000) “waiver” required extensive training and registration, including an 8 h training session for physicians, and 24 h of training for Advanced Practice Practitioners. An X-Waiver is no longer required to prescribe buprenorphine for OUD. Thus, providers with DEA registration for schedule III medications may prescribe buprenorphine as permitted by state law. Patient caps were also eliminated on the number of patients that may be treated with buprenorphine [27]. Buprenorphine for chronic pain was already exempt from the requirement for an X-waiver. Providers might not be aware of current regulations.

### 7.2. Common Misconceptions Surrounding Buprenorphine

*Common* misconceptions surrounding buprenorphine include: (1) “Partial” agonist has been misinterpreted to imply “less effective” analgesia, (2) a lack of awareness of conversion strategies and fear of precipitating withdrawal when transitioning from a full MOR agonist to buprenorphine has hindered use, (3) the stigma of buprenorphine as a treatment for OUD might prevent patients and providers from embracing its use as an analgesic [8].

## 8. Palliative Care and Pharmacy Collaboration

Prescribers and pharmacists need to collaborate to optimize the use of buprenorphine. Open channels of communication help facilitate improved patient care. There is a shared responsibility between prescribers and pharmacists to properly prescribe and dispense controlled substances, as well as to avoid drug–drug interactions and consider the safest and most effective treatment options. The pharmacist might need to contact the prescriber to clarify and consider patient needs. This includes when and where the patient will obtain the buprenorphine and ensuring prior authorizations are completed correctly. Pharmacists play a critical role in decreasing overdose deaths, decreasing overall mortality, decreasing illicit opioid use and improving medication adherence. Given that buprenorphine is very rarely associated with overdose mortality it is an excellent agent for pharmacists to discuss with prescribers. Pharmacists educating other pharmacists is a critical step to make widespread use of buprenorphine a reality [28].

An additional area of importance is that some States in the United States of America are considering legislature that would allow pharmacists to prescribe buprenorphine in the setting of treating opioid use disorder following the passage of the Mainstreaming Addiction Treatment (MAT) Act [29,30].

## 9. Conclusions

Buprenorphine is an attractive opioid for many patients with pain, including those in palliative care or hospices. Its complex activity in multiple opioid receptor subtypes offers a multitude of advantages over traditional opioids without compromising its analgesic benefits. The safety profile of buprenorphine includes a ceiling effect on respiratory depression, less constipation, less depression, and limiting tolerance. The multiple formulations allow for use to be tailored to patients’ needs. Misconceptions and stigma surrounding the use of buprenorphine for OUD should not prevent providers from initiating therapy.

## Figures and Tables

**Table 1 pharmacy-12-00078-t001:** Advantages of buprenorphine unique beta-arrestin and KOR binding.

Results of Less Beta-Arrestin Signaling
Less constipation
Less respiratory depression (“ceiling effect” on respiratory depression)
Less physical dependence
Less tolerance
Less MOR internalization and down-regulation (endocytosis)
**Results of Inverse KOR Agonist**
Anxiolytic effect
Antidepressant effect
Reduced suicidal tendencies
Reduced opioid cravings
Reduced stress-like effect
Reduced addictive potential

**Table 2 pharmacy-12-00078-t002:** Buprenorphine formulations.

Formulations	Brand Names (in US)	Dosage Strength and Frequency	Generic Available	Bioavailability	Time to Peak	Half-Life Elimination
A. For Pain Management						
Injection solution (IM or IV)	Buprenex	0.3 mg/mL Every 6–8 h as needed	Yes	70%	1 h	1.2–7.2 h (IV)
Buccal film	Belbuca	75 mcg 150 mcg 300 mcg 450 mcg 600 mcg 750 mcg 900 mcg Every 12 h	No	46–65%	2.5–3 h	27.6 ± 11.2 h
Transdermal Patch	Butrans	5 mcg/h 7.5 mcg/h 10 mcg/h 15 mcg/h 20 mcg/h Every 7 days	Yes	15%	17 h	26 h
B. For Opioid Use Disorder						
Subcutaneous prefilled syringe	Sublocade	100 mg/0.5 mL 300 mg/1.5 mL Every month	No	30–40%	24 h	43 to 60 days
Subcutaneous prefilled syringe	Brixadi Brixadi weekly	8 mg 16 mg 24 mg 52 mg 64 mg 96 mg 128 mg Every week or month	No	30–40%	6–10 h (monthly) 24 h (weekly)	3 to 5 days (weekly Brixadi), 19 to 26 days (monthly Brixadi)
Sublingual tablet	Subutex	2 mg 8 mg	Yes	29%	30 min to 1 h	37 h
Buprenorphine+ Naloxone Buccal and Sublingual Film and Tablets	Suboxone Zubsov	2 mg ^a^ 4 mg 8 mg 12 mg	Yes	Variable	20–60 min	24–42 h ^b^

^a^ selected list. ^b^ Suboxone.

**Table 3 pharmacy-12-00078-t003:** Common CYP3A4 inhibitors and inducers.

Common CYP3A4 Inhibitors [17]	Common CYP3A4 Inducers [17]
Ketonconazole	Phenytoin
Clarithromycin	Carbamazepine
Ritonavir	Rifampicin
Itraconazole	St. John’s Wort
Telithromycin	Barbiturates

**Table 4 pharmacy-12-00078-t004:** Direct conversion strategy to switch to buprenorphine from traditional opioids.

Patients Taking ≤ 90 MME	Patients Taking > 90 MME
1. Stop the current opioid after nighttime dose 2. Add α2 agonist or immediate-release opioid 3. Start buprenorphine in the morning according to prescribing directions (10 mcg/h patch or 150 mcg buccal film twice daily) 4. Titrate as needed for pain relief	1. Stop the current opioid after nighttime dose 2. Add α2 agonist or immediate-release opioid 3. Start buprenorphine in the morning according to prescribing directions (20 mcg/h patch or 300 mcg buccal film twice daily) 4. Titrate as needed for pain relief

**Table 5 pharmacy-12-00078-t005:** Patients for whom buprenorphine might be a safer alternative.

Older adults
Renal impairment
Concomitant use of gabapentinoid or benzodiazepine
Pre-existing or high risk for substance dependence
Need for long-term opioid use
Patients at high risk for respiratory depression: COPD and OSA
Patients not tolerating other opioids
Chronic non-cancer pain (e.g., post-herpetic neuralgia and osteomyelitis)
Patients with dysphagia who require a long-acting pain alternative

COPD: chronic obstructive pulmonary disease; OSA: obstructive sleep apnea.

## Data Availability

Data are contained within the article.

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
