# Peer review of "Buprenorphine Use for Analgesia in Palliative Care"

_pharmacy, 2024, doi:10.3390/pharmacy12030078_

Round 1

Reviewer 1 Report

Comments and Suggestions for Authors

Thank you for an interesting case and review

Author Response

We thank the reviewer for their time in reviewing our manuscript.

Reviewer 2 Report

Comments and Suggestions for Authors

The article is very much written with a biased view and contains several misconceptions and firm statements without substantiation.

-          “There is also less drug interaction with buprenorphine which is metabolized via CYP3A4 102 compared with methadone for which CYP2B6 is the principal determinant of metabolism, clearance and elimination” is not right: of all medications 44% is metabolized via CYP 3a4 and only 25% via CYP 2D6 so buprenorphine has in theory more interactions.

-          “Cannabis can increase serum buprenorphine levels and thus patients should be counseled on the risk of withdrawal symptoms if cannabis is stopped” strange reference because this is an unsubstantiated or referenced sentence from a non-research article.

-          “Absorption of the drug may be increased when a patient has elevated body temperature” speculation: no evidence

-          “buprenorphine through the buccal and sublingual formulations are subjected to first pass metabolism”:  not true: transmucosal delivery is exempted from first pass

-          “Additionally, when a CYP3A4 inhibitor or inducer is administered concurrently with bupren- orphine, the drug level of buprenorphine may change, and additional monitoring of liver 158 function tests will be warranted.”: nonsense advice: you do not dose on guided serum levels, but on effect in the patient

-          “Buprenorphine is typically not listed in the Morphine….” This is only true for US

-          “Older adults commonly have impaired renal function; thus all opioids, except buprenorphine..”: not true! Only methadone and fentanyl are safe: and buprenorphine has an active metabolite

-          “In older adults, already at risk of immunosenescence” is that true??

-          “It has also been suggested that buprenorphine might be beneficial in neuropathic pain” this sentence comes from a consensus statement without reference!

-          Former Regulatory…. Only of importance for US

-          “Palliative care and collaboration: superfluous

Author Response

Response to reviewer#2:  Buprenorphine Use for Analgesia in Palliative Care

We thank the reviewer for their time in reviewing our manuscript and attention to detail. We have revised our manuscript taking all their excellent points into consideration. 

  1. “There is also less drug interaction with buprenorphine which is metabolized via CYP3A4 compared with methadone for which CYP2B6 is the principal determinant of metabolism, clearance and elimination”

The text has been revised to include all the CYP metabolites.  A supporting reference has been added.

  1. “Cannabis can increase serum buprenorphine levels and thus patients should be counseled on the risk of withdrawal symptoms if cannabis is stopped [5].”

To enhance the clarity of this statement we have added a paragraph highlighting the dangers of intoxication from new cannabis use, as well as an additional reference supporting the risk of elevated buprenorphine and norbuprenorphine levels in cannabis-users, likely due to CYP3A4, as well as the lower levels of metabolite-to-parent drug ratio in cannabis-users.

  1. “Absorption of the drug may be increased when a patient has elevated body temperature or applying a heating pad onto the transdermal system, reports have shown blood concentrations of buprenorphine increased by 26% to 55% when heating pad is used [9, 10, 11].”

The statement was supported by the Thomas reference (now #12): “In Vitro-In Vivo Correlation of Buprenorphine Transdermal Systems Under Normal and Elevated Skin Temperature. Pharmaceutical Research”  We have added an additional explanation from that paper describing the results from the in vitro permeation tests.

  1. “Individuals who received buprenorphine through the buccal and sublingual formulations are subjected to first pass metabolism, however, those who take parenteral formulations have been reported to experience less drug-drug interactions associated with CYP3A4 in-hibitors or inducers [12].”

We apologize for the confusion. Although sublingual formulations bypass the first pass metabolism, study has shown a significant portion of sublingual buprenorphine is swallowed, as evidenced by the presence of norbuphrenorphine (the active metabolite) in the systemic circulation. The sentence is rephrased to be clearer to readers.

  1. “Additionally, when a CYP3A4 inhibitor or inducer is administered concurrently with bu-prenorphine, the drug level of buprenorphine may change, and additional monitoring of liver function tests will be warranted.”

We have modified this sentence to include monitoring of efficacy and added the supporting reference.

  1. “Buprenorphine is typically not listed in the Morphine milligram equivalent (MME) table in the clinical practice guideline and opioid equivalency table [15].”

The authors are not aware of any opioid equivalency tablets that list buprenorphine.  This statement is supported by references 17 and 18.  We have added a direct quote from the CDC Clinical Guideline:  “Buprenorphine should not be counted in the total MME/day in calculations because of its partial agonist properties at opioid receptors that confer a ceiling effect on respiratory depression.

We have also added the reference from the Health and Human Services 2018 table (website most recently updated 8/25/2020) that also does not include a MME for buprenorphine, now #19.

Table 4 includes an expert panel suggestion for direct conversion strategies (reference 20). 

  1. “Older adults commonly have impaired renal function; thus all opioids, except buprenorphine, would require longer time interval, reduced doses, and closer monitoring of creatinine clearance. “

This sentence is supported by the guideline provided in #24:  Pergolizzi J, Böger RH, Budd K, et al. Opioids and the management of chronic severe pain in the elderly: consensus statement of an International Expert Panel with focus on the six clinically most often used World Health Organization Step III opioids (buprenorphine, fentanyl, hydromorphone, methadone, morphine, oxycodone). Pain Pract. 2008;8(4):287-313. doi:10.1111/j.1533-2500.2008.00204.x”   We have added the reference immediately following the statement to enhance the clarity of the source.

  1. “In older adults, already at risk of immunosenescence” is that true??

Thank you for suggestion to expand upon this statement.  The following reference has been added to support this statement, as well as a more detailed explanation of the process of immunosenescence. “Lee K, Flores RR, Jang IH, Saathoff A, Robbins PD. Immune Senescence, Immunosenescence and Aging. Front Aging. 2022; 3: 900028.  https://www.ncbi.nlm.nih.gov/pmc/articles/PMC9261375/

doi: 10.3389/fragi.2022.900028  PMCID: PMC9261375  PMID: 35821850

  1. “It has also been suggested that buprenorphine might be beneficial in neuropathic pain” this sentence comes from a consensus statement without reference!

We regret not having added enough supporting documentation.  The following references have been added to support use of buprenorphine use in neuropathic pain:

Minerva Anestesiol   . 2013 Aug;79(8):871-83.    Epub 2013 Apr 5.

Safety and efficacy of transdermal buprenorphine and transdermal fentanyl in the treatment of neuropathic pain in AIDS patients

A CannetiM LuziP Di MarcoF CannataF PasqualittoA SpinoglioC Reale

In addition we have added a sentence with accompanying references suggesting the rationale for using buprenorphine in this setting, as well as animal studies supporting its use.  

          Former Regulatory…. Only of importance for US

It is indeed true that this is true to providers in the US.  The authors felt it relevant to include the former regulations in the United States because the changes occurred in 12/2022 and thus there might be providers in the United States who might not yet be aware of these changes. We have added the “in the United States” to the Former Regulatory Barrier section to alert readers for whom this would be pertinent. 

-          “Palliative care and collaboration: superfluous

This paragraph was added in an attempt to expand upon the importance of collaboration between pharmacists and palliative care providers and to highlight the value of this article in a Pharmacy Journal.

 The authors thank the reviewer for their excellent suggestions to improve upon this paper.

Sincerely,

Dr Leanne Jackson

Reviewer 3 Report

Comments and Suggestions for Authors

Thank you for the opportunity to review this article, which covers the use of buprenorphine in Palliative Care. Overall, this is a well-written and pertinent article that will be of interest to the Palliative Care community, in which buprenorphine is not as commonly used. I very much appreciated Table 4, which makes this article particularly applicable for physicians. 

1. Line 187 - is the "stop-start" method also referring to the direct conversion strategy above? It would be helpful to make it explicit what is recommended for patients with chronic pain (not OUD). 

Author Response

Response to reviewer#3:  Buprenorphine Use for Analgesia in Palliative Care

We thank the reviewer for their time in reviewing our manuscript and attention to detail. We have revised our manuscript taking their excellent point into consideration. 

  1. Line 187 - is the "stop-start" method also referring to the direct conversion strategy above? It would be helpful to make it explicit what is recommended for patients with chronic pain (not OUD). 

We have added a statement directing the reader to the microdosing strategies or expert panel recommendations in table 4 (which the reviewer found helpful) and noted that this is the recommend strategy for the palliative care population or for patients with chronic pain who need a safer option for analgesia.

Round 2

Reviewer 2 Report

Comments and Suggestions for Authors

-          I don’t think my first objection is answered satisfactory. The authors expanded on the CYP enzymes involved in the metabolism of methadone but failed to demonstrate that buprenorphine has less interactions.

-          I’m sorry I overlooked next point last time:

“buprenorphine can be titrated to a stable dose within several days as opposed 105 to several weeks to months.” This is not true: you reach the steady state after 4.5 x the half-life: for buprenorphine that’ll be 5 days (patch), for methadone 5 – 7 days.

-          The authors are not aware of any opioid equivalency tablets that list buprenorphine:

Dutch National guideline: Opioidrotatie - Richtlijnen Palliatieve zorg (palliaweb.nl) I don’t agree with the statement about buprenorphine and calculations: yes, there is a ceiling effect for respiratory depression but the painkilling properties are as wit hall other opioids: of course you should count it in.

Dutch evidence based guideline: Palliatieve zorg bij eindstadium nierfalen - Pijn - Richtlijn - Richtlijnendatabase:

-          Opioids and kidneyfunction:

-           If paracetamol has insufficient effect, start with a strong-acting opioid.

-          Fentanyl transdermal (at a dose of 12 ug/hour) is the drug of first choice.

-          Slow-release hydromorphone, slow-release oxycodone, tramadol and buprenorphine transdermal are alternatives. In doing so, start with the lowest dose and increase cautiously.

-          Methadone should only be prescribed by or in consultation with someone experienced with this drug.

-          The use of codeine and morphine is not recommended.

-          and

-          Opioids in renal failure and dialysis patients - Journal of Pain and Symptom Management (jpsmjournal.com)

-          You cannot state that buprenorphine is the safest with respect to the immune system. You reference says: “In summary, there is a reciprocal interaction between the immune system and endogenous as well as exogenous opioids. Further to the existing epidemiological studies, controlled clinical studies are needed in the future to elucidate the role of the opioid-immune system interaction in patients and to determine its clinical relevance.” So it can be good or bad and nobody knows whether this is clinically relevant.

-          Neuropathic pain and buprenorphine: Your reference 1 states: buprenorphine and fentanyl are equally effective and Cochrane: There was insufficient evidence to support or refute the suggestion that buprenorphine has any efficacy in any neuropathic pain condition. Buprenorphine for neuropathic pain in adults - PMC (nih.gov)

Author Response

The authors again thank the reviewer for their detailed review of our manuscript. We have revised the manuscript taking all their excellent points into consideration.     

-   I don’t think my first objection is answered satisfactory. The authors expanded on the CYP enzymes involved in the metabolism of methadone but failed to demonstrate that buprenorphine has less interactions.

To address this concern, we have reworded the sentence to, “Buprenorphine is metabolized via CYP3A4 and methadone is metabolized through CYP3A4, CYP2B6 is and CYP2D6 [5,6].”

-          I’m sorry I overlooked next point last time:

“buprenorphine can be titrated to a stable dose within several days as opposed 105 to several weeks to months.” This is not true: you reach the steady state after 4.5 x the half-life: for buprenorphine that’ll be 5 days (patch), for methadone 5 – 7 days.

Thank you for your comment. We agree that this sentence needs clarification. We modified this sentence to, “The half-lives of IM, buccal, and transdermal buprenorphine are 2 to 3 hours, 27 hours, and 26 hours respectively.”

-          The authors are not aware of any opioid equivalency tablets that list buprenorphine:

Dutch National guideline: Opioidrotatie - Richtlijnen Palliatieve zorg (palliaweb.nl) I don’t agree with the statement about buprenorphine and calculations: yes, there is a ceiling effect for respiratory depression but the painkilling properties are as wit hall other opioids: of course you should count it in.

Dutch evidence based guideline: Palliatieve zorg bij eindstadium nierfalen - Pijn - Richtlijn - Richtlijnendatabase:

-          Opioids and kidneyfunction:

-           If paracetamol has insufficient effect, start with a strong-acting opioid.

-          Fentanyl transdermal (at a dose of 12 ug/hour) is the drug of first choice.

-          Slow-release hydromorphone, slow-release oxycodone, tramadol and buprenorphine transdermal are alternatives. In doing so, start with the lowest dose and increase cautiously.

-          Methadone should only be prescribed by or in consultation with someone experienced with this drug.

-          The use of codeine and morphine is not recommended.

The Dutch guidelines, referring to transdermal buprenorphine 52mcg (OME: 120mg) and 105mcg (OME: 240mg) have been included into the article. 

-          and

-          Opioids in renal failure and dialysis patients - Journal of Pain and Symptom Management (jpsmjournal.com)

The authors have read through the above article on “Opioids in Renal Failure and Dialysis Patients” in JPSM and do not see buprenorphine mentioned in this article. 

 -          You cannot state that buprenorphine is the safest with respect to the immune system. You reference says: “In summary, there is a reciprocal interaction between the immune system and endogenous as well as exogenous opioids. Further to the existing epidemiological studies, controlled clinical studies are needed in the future to elucidate the role of the opioid-immune system interaction in patients and to determine its clinical relevance.” So it can be good or bad and nobody knows whether this is clinically relevant.

 Statements about buprenorphine being the safest for impaired immune have been removed.

-          Neuropathic pain and buprenorphine: Your reference 1 states: buprenorphine and fentanyl are equally effective and Cochrane: There was insufficient evidence to support or refute the suggestion that buprenorphine has any efficacy in any neuropathic pain condition. Buprenorphine for neuropathic pain in adults - PMC (nih.gov)

Sentences about use of buprenorphine for neuropathic pain have been removed.